Circadian preferences of birdwatchers in Poland: do “owls” prefer watching night birds, and “larks” prefer daytime ones?

Frątczak Martyna 1
http://orcid.org/0000-0003-4382-7051 Sparks Tim H. 1
http://orcid.org/0000-0002-7357-2793 Randler Christoph 2
http://orcid.org/0000-0002-8358-0797 Tryjanowski Piotr 1 piotr.tryjanowski@gmail.com
1 Poznan University of Life Sciences , Poznan , Poland
2 Eberhard-Karls-Universität Tübingen , Tubingen , Germany
Wink Michael
Electronic publication date: 2020 Mar 2
Publication date: 2020
Volume: 8
Electronic Location ID: e8673
Received 2019 May 31; Accepted 2020 Jan 31
Copyright: © 2020 Frątczak et al.
Copyright year: 2020
Copyright holder: Frątczak et al.
License: This is an open access article distributed under the terms of the Creative Commons Attribution License, which permits unrestricted use, distribution, reproduction and adaptation in any medium and for any purpose provided that it is properly attributed. For attribution, the original author(s), title, publication source (PeerJ) and either DOI or URL of the article must be cited.
License URL: https://creativecommons.org/licenses/by/4.0/

Keywords: Circadian preference, Environmental attitudes, Morningness–eveningness, Birdwatching

Funding: The authors received no funding for this work.

==============================
Birdwatching is a very popular and increasing leisure activity, and the study and observation of birds is more popular in the morning due to the greater activity among birds at that time of day. The aim of our study was to find out whether there was a relationship between the circadian preference of observers and their favourite bird species and whether it was influenced by such factors as professional status, age and gender. In an e-mail survey we asked a total of 433 Polish ornithologists (professionals) or birdwatchers (non-professionals) for their morningness–eveningness preferences (four categories) and favourite (open choice) bird species and received 143 responses. The temporal (circadian) preferences of respondents declined from early morning (35.7%) to evening/nighttime (11.4%). Circadian preference categories differed significantly by age, with early morning respondents significantly older. These preference categories did not differ significantly in terms of response time to the survey invitation or in the percentage of their favourite birds that were categorised as daytime birds. A total of 204 species were identified as favourite birds of which 34 species were mentioned by five or more respondents, with only two, the common crane Grus grus and the Eurasian pygmy owl Glaucidium passerinum mentioned by more than 10% of respondents. The white stork Ciconia ciconia was more popular with professionals than non-professionals and the swift Apus apus less popular. A significant gender × circadian preference interaction was detected for the percentage of favourite birds categorised as daytime birds, with fewer daytime birds among early morning female recorders. The presented results are obviously of a correlative nature, but open the door for further, more advanced study and suggest there may be a need to investigate temporal biases when analysing citizen-based data.

Introduction

Many factors influence human attitudes to animals, including animal traits such as colour, size, rarity and popularity and human traits such as gender, age, education level and even food preferences (reviewed in: Prokop & Randler, 2018). Obviously, both animal and human perception have co-evolved, and the latter changes due to life experience and education (Prokop, Kubiatko & Fančovičová, 2008; Prokop & Randler, 2018). Recently, dealings with animals have not necessarily been linked just to a targeted use (e.g. food source, guard role, beast of burden), but simply watching animals for fun, of which birdwatching is one example, is increasing worldwide (Kronenberg, 2016; Callaghan et al., 2018). Bird species differ in many ecological traits and these can also affect human behaviour, for example searching for information on particular species (Żmihorski et al., 2013; Correia et al., 2016, 2017) and even willingness to pay for their conservation (Raab, Randler & Bogner, 2018). However, people watching birds, and other taxa, differ in many psychological and physical traits and one very marked behavioural difference is their particular morningness–eveningness spectrum of activity (Werner, Geisler & Randler, 2015). In the case of bird observation, this can affect not only intellectual and physical peak performance during the day (Pica et al., 2015; Randler, Faßl & Kalb, 2017; Arrona-Palacios & Díaz-Morales, 2018), but also punctuality (Werner, Geisler & Randler, 2015) and risk-taking (Wang & Chartrand, 2015), which can be useful in finding and watching rare birds, which is a particularly focussed activity of some birdwatchers (Żmihorski, Sparks & Tryjanowski, 2012; Callaghan et al., 2018).

The morningness–eveningness preference may change in relation to age, gender and even seasons (Pica et al., 2015; Randler, Faßl & Kalb, 2017), which also makes this interesting to study in terms of the potential relationship between humans and birds. Birds are a very variable taxonomical group. There are more than 10,000 species, they live in different habitats and they have different seasonal and daily activities mainly related to breeding, wintering and seasonal migrations (Del Hoyo, Elliott & Sargatal, 1992–2011; Jetz et al., 2012), all of which combine to strongly affect people who watch and study birds (Callaghan et al., 2018). Because people have different financial resources, time availability, personal preferences, skills, knowledge and commitment, they may focus on different bird species/taxonomic groups (Lee & Scott, 2004; Prokop, Kubiatko & Fančovičová, 2008; Yarwood, Weston & Garnett, 2014). Furthermore, the psychological terminology on circadian preference and morningness–eveningness has always been linked to bird behaviour, as shown by use of the classical phrase ‘owls vs. larks’ (Pica et al., 2015; Putilov, Donskaya & Verevkin, 2015). Furthermore, differences in daily patterns of data collection and a focus on particular bird species may attach a bias to obtained results. Given the current debate on the importance of data quality (La Sorte & Somveille, 2020) this is a very pertinent issue, also in the context of bird monitoring schemes (Snäll et al., 2011).

However, we do not know of a single study that has linked morningness–eveningness to preferences of particular bird species or groups and therefore it is difficult to specify exact testable hypotheses. However, we hazard a guess that people with stronger morningness than eveningness will prefer morning rather than nocturnally active bird species and vice versa. Therefore, based on an e-mail questionnaire to birdwatchers, we attempted to answer whether: (1) if there is a relationship between observer morningness–eveningness and favourite bird species; (2) if there are differences between amateur birdwatchers and professional ornithologists in this human trait; (3) if potential differences are influenced by other factors (age and gender).

Methods

Data collection

At 00:00 h on the 19 February 2019 the first author (MF) sent an e-mail questionnaire to 433 (356 male, 77 female) Polish ornithologists and advanced birdwatchers and asked for a response by e-mail. Potential participants were chosen by preparing a database of e-mail addresses using information provided in scientific publications, including short notes published in Polish naturalists’ journals dedicated to birds. We accessed 28 journals publishing even local records of rare species, unusual behaviour, species lists and even notes on untypical plumage (mainly in Polish. The whole list of accessed journals is available in Supplemental Material). Participants were asked for up to five favourite bird species (without any suggestion as to local or exotic), their year of birth and their classical circadian general type as used in similar studies (Randler, Schredl & Göritz, 2017) as follows: (A) definitely morning type; (B) more morning than evening type; (C) more evening than morning type; and (D) definitely evening and night type. Measuring circadian preference in this way could be criticised because it is based on a single question, but Loureiro & Garcia-Marques (2015) showed that results were rather similar, whether based on only a single question or derived from the full scale. Randler, Schredl & Göritz (2017) showed that results were comparable, irrespective of whether a four-item version or a single-item measurement was used; both measures of chronotype revealed identical results. This is also in line with Loureiro & Garcia-Marques (2015). We closed the survey two weeks after the initial e-mail posting. For each respondent we recorded when their reply was sent and used this to calculate the time to respond, expressed in decimal hours.

Statement on ethical approval

Research reported in this manuscript has been conducted in an ethical and responsible manner, in full compliance with all relevant codes of experimentation and legislation.

Data analyses

For further analyses we added information on gender and we divided participants, according to their academic degrees and jobs, into two general categories: professional and non-professional (amateurs). The suggested favourite bird species were divided into two broad categories–nighttime birds (such as owls, rails etc.) and daytime birds (the majority of passerines, cranes, ducks etc.) using data from handbooks (mainly Del Hoyo, Elliott & Sargatal, 1992–2011). For each respondent the percentage of favourite birds that were daytime birds was calculated.

Because of the rather skew nature of some of the variables it was decided to focus on nonparametric tests. Kruskal Wallis tests, adjusted for ties, were used to compare scale variables to categories and chi squared tests to compare categories with one another. Where expected numbers were so low as to question the validity of chi squared results, Fisher exact tests were used. For testing individual species such tests were restricted to species mentioned by 10 or more respondents. Spearman correlations were used to compare scale variables. Binary logistic regression was used to compare the proportions of favourite birds identified as daytime birds to year of birth, professional status, gender and time preference category. The model initially included all main effects and two-way interactions, but nonsignificant two-way interactions were dropped one-by-one until only the main effects and any significant interactions remained.

Year of birth was used in analyses, but is presented as age in results. The significance threshold in all analyses was taken as P = 0.05. All analysis was carried out in Minitab 18 and SPSS 24.

Results

Description of participants

Complete or near complete responses were received from 143 individuals (33.0%), with response times ranging from 3 min to 14 days (the cut-off point). Response rates were 31.7% for males and 39.0% for females with no significant difference between the two (χ12=0.73, P = 0.394).

Response times were positively skewed with a median of 10 h and a mean of 44 h. Years of birth were provided by 140 individuals, indicating a mean age of 45 years (range 22–85). The professional status of the 142 birders providing detail was roughly evenly split between non-professionals (51.4%) and professionals (48.6%). As expected from the targeted individuals, the majority of respondents (79%) were male. The circadian preferences of respondents changed from early morning; 50 (35.7%) coded A, 41 (29.3%) coded B, 33 (23.6%) coded C and 16 (11.4%) coded D, 3 people didn’t provide a response to this question.

It is worth noting that 11 complaints (mainly about the aims and methods of the study) were also received, which in combination with other responses indicate a response rate of 35.6% of the targeted population. Nine of the complaints came from professional birders and a chi squared test indeed confirmed that professionals were significantly more likely to complain (χ12=4.70, P = 0.030). There was no significant association between gender and complaining (χ12=1.18, P = 0.277). Furthermore, those complaining were significantly older (H = 11.91, P = 0.001, medians 64, 42 years).

Professional birdwatchers were significantly older (H = 5.89, P = 0.015) than non-professionals (median ages 47 and 41 years respectively), but did not differ significantly in response times to the survey (H = 0.72, P = 0.395) or in the percentage of their favourite birds that were categorised as daytime birds (H = 0.29, P = 0.589). There was no significant association between professional status and gender (χ12=0.76, P = 0.384) or between professional status and circadian preference (χ32=5.34, P = 0.149).

Female respondents were significantly younger than male respondents (H = 11.91, P = 0.001, medians of 37 and 45 years respectively), but did not differ significantly in terms of response times (H = 0.04, P = 0.835) or percentage of their favourite birds that were categorised as daytime birds (H = 0.12, P = 0.726). There was a suggestion that females were less likely to be category A recorders but this did not quite reach significance (Fisher exact test P = 0.053).

Circadian preference categories differed significantly in age, with category A respondents noticeably older (H = 10.56, P = 0.014, medians 48, 41, 40 and 42 years respectively for categories A–D). Circadian preference categories did not differ significantly in terms of response time (H = 3.29, P = 0.349) or percentage of their favourite birds that were categorised as daytime birds (H = 2.29, P = 0.514).

Favourite bird species

A total of 204 species were identified as favourite birds, of which 184 (90.2%) were classified as daytime birds. A full table of species indicating their daytime/nighttime category is given in Supplemental Material 1. The 34 species mentioned by five or more respondents are listed in Table 1. Only two species, the common crane Grus Grus and the Eurasian pygmy owl Glaucidium passerinum, were mentioned by more than 10% of respondents. Only the 12 species that were mentioned 10 or more times were subject to further analysis. This additional analysis suggested that the white stork Ciconia ciconia was more popular with professionals than non-professionals (13% cf. 3%) and the swift Apus apus less popular (3% cf. 12%). No significant gender preferences were detected in this subset of species. There was a significant difference in the circadian preference categories for swift which was not mentioned by any of the category A respondents.

Table 1 The 34 most popular species (those with n ≥ 5) reported by respondents.

In addition to the overall number and percentages, percentages are also given by professional status (NP nonprofessional, P professional), gender (F female, M male) and circadian preference (categories A–D, see main text). Significance (indicated by *) tested using chi squared or Fisher exact tests for the 12 species with n ≥ 10. Species in grey shading were categorised as nighttime; all others as daytime.

Species	n	%	NP%	P%		F%	M%	A%	B%	C%	D%		
Grus grus	28	19.6	23.3	15.9		23.3	18.6	20.0	29.3	15.2	6.3		
Glaucidium passerinum	16	11.2	12.3	10.1		10.0	11.5	10.0	14.6	9.1	12.5		
Alcedo atthis	14	9.8	11.0	8.7		10.0	9.7	12.0	7.3	9.1	12.5		
Upupa epops	14	9.8	11.0	8.7		10.0	9.7	8.0	12.2	12.1	6.3		
Turdus merula	13	9.1	11.0	7.2		6.7	9.7	14.0	2.4	9.1	12.5		
Aegithalos caudatus	12	8.4	8.2	8.7		16.7	6.2	8.0	9.8	9.1	6.3		
Ciconia ciconia	11	7.7	2.7	13.0	*	3.3	8.8	12.0	4.9	9.1	0.0		
Vanellus vanellus	11	7.7	8.2	7.2		6.7	8.0	2.0	9.8	9.1	18.8		
Apus apus	11	7.7	12.3	2.9	*	10.0	7.1	0.0	14.6	12.1	6.3	*	
Merops apiaster	11	7.7	9.6	5.8		10.0	7.1	10.0	2.4	12.1	0.0		
Haliaetus albicilla	10	7.0	8.2	5.8		6.7	7.1	8.0	9.8	0.0	6.3		
Lanius collurio	10	7.0	5.5	8.7		6.7	7.1	10.0	2.4	9.1	6.3		
Strix aluco	9	6.3	9.6	2.9		6.7	6.2	6.0	12.2	3.0	0.0		
Circus pygargus	9	6.3	8.2	4.3		6.7	6.2	6.0	9.8	3.0	6.3		
Corvus monedula	9	6.3	2.7	10.1		10.0	5.3	8.0	7.3	6.1	0.0		
Ciconia nigra	8	5.6	8.2	2.9		6.7	5.3	6.0	7.3	3.0	6.3		
Corvus frugilegus	8	5.6	4.1	7.2		6.7	5.3	10.0	0.0	6.1	6.3		
Corus corax	8	5.6	8.2	2.9		13.3	3.5	4.0	7.3	0.0	12.5		
Galerida cristata	8	5.6	6.8	4.3		3.3	6.2	8.0	4.9	3.0	6.3		
Erithacus rubecula	7	4.9	6.8	2.9		3.3	5.3	8.0	4.9	3.0	0.0		
Bucephala clangula	7	4.9	4.1	5.8		3.3	5.3	6.0	2.4	6.1	6.3		
Sturnus vulgaris	6	4.2	5.5	2.9		3.3	4.4	8.0	0.0	6.1	0.0		
Parus major	6	4.2	1.4	7.2		3.3	4.4	6.0	0.0	6.1	6.3		
Lyrurus tetrix	6	4.2	5.5	2.9		0.0	5.3	2.0	7.3	6.1	0.0		
Athene noctua	6	4.2	4.1	4.3		10.0	2.7	0.0	0.0	9.1	18.8		
Aquila chrysaetos	6	4.2	8.2	0.0		6.7	3.5	2.0	9.8	0.0	6.3		
Limosa limosa	6	4.2	5.5	2.9		3.3	4.4	2.0	7.3	3.0	6.3		
Tyto alba	5	3.5	4.1	2.9		0.0	4.4	6.0	0.0	6.1	0.0		
Milvus milvus	5	3.5	4.1	2.9		3.3	3.5	6.0	0.0	3.0	6.3		
Tetrao urogallus	5	3.5	2.7	4.3		3.3	3.5	0.0	7.3	6.1	0.0		
Sylvia nisoria	5	3.5	2.7	4.3		0.0	4.4	2.0	4.9	6.1	0.0		
Coracias garrulus	5	3.5	2.7	4.3		3.3	3.5	4.0	2.4	0.0	12.5		
Strix nebulosa	5	3.5	2.7	4.3		6.7	2.7	4.0	0.0	3.0	12.5		
Clanga pomarina	5	3.5	6.8	0.0		3.3	3.5	4.0	2.4	0.0	12.5		

There were no significant Spearman correlations between age, response times and percentage of favourite birds that were categorised as daytime birds (all P > 0.297).

Binary logistic regression on the proportion of favourite birds categorised as daytime produced an overall significant model (χ92=23.47, P = 0.005) with one significant main effect (gender: χ12=5.09, P = 0.024) and one significant interaction (gender × circadian preference: χ32=14.26, P = 0.003). In addition, circadian preference was very close to significance (χ32=7.80, P = 0.0503). This interaction is shown in Fig. 1 where female respondents in category A reported fewer favourite daytime birds than their male counterparts and in contrast to the pattern in other categories.

Figure 1 The mean percentage ±SE of favourite birds classified as daytime by gender and circadian preference.

Discussion

The number of participants may appear lower than expected in western European countries such as the Netherlands, Germany and especially the UK (Cocker, 2012). But in reality we asked a large proportion of bird observers that usually participate in national citizen science projects in Poland, such as Common Birds Monitoring Scheme, Winter Bird Counts and BirdFeeder Project. These schemes usually attract only 100–180 people. We used an e-mail based questionnaire, with an overall response rate of 35.6% which may appear low, but is similar to the average rate in other studies using similar methods in psychology (Cook, Heath & Thompson, 2000; Sheehan, 2001). We also noted with great interest the few complaining e-mails where respondents, mainly professionals, expressed concerns over the aims and methodology of the study. It may seem obvious that a significantly greater rate of complaint would derive from professional ornithologists rather than amateur birdwatchers, because professionals focus much more on science than simply gaining pleasure from watching birds (Yarwood, Weston & Garnett, 2014; Yarwood, Weston & Symonds, 2019 and see also below).

The study also suggests that watching birds is still more popular among men than women, similar to other studies in Poland, but which were based on a much smaller sample size (Sklodowski & Jurkowska, 2015). However, it is interesting to note that women participating in our study were significantly younger than men, which is a sign of an encouraging (probably global) trend that birdwatching has recently become more popular with women (Cooper & Smith, 2010). The most represented circadian preference was the ‘early morning type,’ a finding similar to that found by Raab, Randler & Bogner (2018). Moreover, those authors even suggested that students with a morningness preference were more likely to be protective of and appreciative towards, nature, including birds. Furthermore, early morning (lark type) people more often participated in voluntary activities, shared information, were open to new ideas, were faster and more punctual (Song & Stough, 2000; Werner, Geisler & Randler, 2015). On the other hand, if we compare birders with the general Polish population (Jankowski, 2013; Tęgowska, Sobocińska & Maliszewska, 2014) there is still apparent a visible bias to the early morning type. In a correlative-type study like ours it is impossible to determine if early morning types are more likely to take up an ornithological career, or indeed if they just have to get up early to watch more bird species. This can only be resolved if a representative sample of Polish people was surveyed and matched for gender and age, because there are significant age effects in circadian preference in the adult population (Randler et al., 2016).

The 143 respondents mentioned 204 species as favourite birds with great variation between respondents, but some patterns are worth noting. Only two species, the common crane and the Eurasian pygmy owl, were mentioned by more than 10% of respondents. We did not ask why people decided on their particular species, but Eurasian pygmy owl was a surprise to us. The common crane is a more obvious choice because it is a big charismatic bird with a characteristic call, an increasing population size and is also represented in Polish culture and art (Wąs, 2013). Interestingly, the white stork was a more popular choice with professionals than with non-professionals but the swift showed an opposite pattern. The white stork is a Polish icon, well recognised, easy-to-spot even by beginners and very popular in both scientific studies and the cultural life of Poland (Kronenberg, Andersson & Tryjanowski, 2017). There are long term-data on white storks that permit advanced statistical analyses and so it is often chosen as a study species by ornithologists, although it is not very attractive to classical birdwatchers because it is still common, but is more exciting to visitors to Poland rather than local birdwatchers (Kronenberg, 2016; Kronenberg, Andersson & Tryjanowski, 2017). In choosing study species, ornithologists follow different criteria, based on the probability of obtaining a good sample size, conservation status and even size and behaviour (Yarwood, Weston & Garnett, 2014; Yarwood, Weston & Symonds, 2019). After some years of study, they know much about the biology of the species, including its secrets and are fascinated by the species which then also becomes a favourite bird (personal experience of the authors). It is likely that preferences change over time and are sensitive to mass-media information making species more or less charismatic in recent years (Żmihorski, Sparks & Tryjanowski, 2012; Correia et al., 2016, 2017). In our sample, swift is perhaps an indicator of changes in recent times, because the species is an object of recent debate focused on bird conservation in urban areas and information has been promoted on this species biology, ecology and importance to humans (Luniak & Grzeniewski, 2011). It is interesting that this species was not suggested as a favourite bird by any of the ‘early morning’ respondents, which fits especially well with the late afternoon and early evening activity of swifts.

A number of other interesting patterns appear evident in Table 1 but failed to achieve either significance or sufficient sample size for testing. A study based on a larger sample would be needed to confirm or refute any of these apparent patterns. Probably a more advanced study should compare species choice with other factors, for example how much attention she/he pays to that particular bird species, how many publications are dedicated to that particular species and even a measure of emotional attachment. Similar studies could be useful for understanding the choice of different outdoor activities between people, as well as a deeper understanding of the variability of other groups of passionate recreationalists (e.g. anglers, hunters, mountain climbers, cf. Wałęga, Wałęga & Graczyk, 2017). Given that much of the data used to estimate species population sizes and geographical distributions and to produce bird guides, originate from recorders of which our respondents were a sample, it is important to know if their behaviour may subconsciously influence their records. Furthermore, a better understanding of the reasons for the popularity of species could help enormously in planning for and funding, their conservation. One important aspect that chronobiology may contribute to bird research is the influence on citizen science programs. Citizen science programs, such as eBird rely heavily on the submissions of many volunteer birdwatchers (Wiersma, 2010). However, the reporting of given species may be significantly influenced if the chronotype of the birdwatchers and the circadian rhythms of specific species do not map. Early morning bird species may go underreported if most birdwatchers are not morning oriented themselves. In a similar manner, a higher proportion of ‘owls’ in birdwatchers may lead to a better registration of owls in these databases. Of course, these assumptions need further study based on larger samples.

Conclusion

Although much is known about birds and the importance of citizen science, only scarce evidence is available about the nature of birdwatchers, both professional and amateur. These data, especially about their morning or evening preference, may severely influence data collection behaviour but, in this study, we were able to show that there is only a weak influence of circadian preference on leisure birdwatching; thus, we assume a minimal or even absent influence of circadian preference on birdwatching.

Supplemental Information

Supplemental Information 1 Response list by birdwatcher and species.

Click here for additional data file.

Supplemental Information 2 List of journals used to find email addresses of participants. Ordered alphabetically by name.

Click here for additional data file.

Additional Information and Declarations

Competing Interests

Author Contributions

Data Availability

Piotr Tryjanowski is an Academic Editor for PeerJ.

Martyna Frątczak conceived and designed the experiments, performed the experiments, authored or reviewed drafts of the paper, and approved the final draft.

Tim H. Sparks analyzed the data, prepared figures and/or tables, authored or reviewed drafts of the paper, and approved the final draft.

Christoph Randler conceived and designed the experiments, analyzed the data, authored or reviewed drafts of the paper, and approved the final draft.

Piotr Tryjanowski conceived and designed the experiments, analyzed the data, prepared figures and/or tables, authored or reviewed drafts of the paper, and approved the final draft.

The following information was supplied regarding data availability:

Raw data is available as a Supplemental File.

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
