# Peer review of "Circadian preferences of birdwatchers in Poland: do “owls” prefer watching night birds, and “larks” prefer daytime ones?"

_PeerJ, doi:10.7717/peerj.8673_

## Round 0.1 · original submission · Major Revisions

Dear Piotr

Your ms has been reviewed by now. As you can see, both reviewers are not very happy with the ms and have recommended a major revision. If you revise, please follow their advice as we will send the revision to the same reviewers.

Greetings
Michael

Reviewer 1 ·

Basic reporting

see general comments

Experimental design

see general comments

Validity of the findings

see general comments

Additional comments

My main impression is: So what? The study would be well suited as a school-leaving thesis, and the results are in no way exciting. The conclusion ends as usual: "The presented results ... open the door for further, more advanced studies." Why haven't you already done exactly that in this work?
Intr: Many of the statements in the introduction are very vague and superficial. Mixing of professional and non-professional motives and goals.
Meth: Data collection: "At 00:00 hour on the 19 February 2019 the first author sent an e-mail questionnaire to 433 Polish ornithologists - I feel reminded of a sketch by Mr. Bean.
[Annoying and sloppy: In data collection n is 433 - in Summary n=443.]
Pseudo-correctness: l 114-119.
Results: only 32% of the questionnaires were returned. Why so few? Did the respondents find the questionnaire too lapidary? - Eleven complaints: Where is the significance of the analysis (icl. statistics)? - favourite bird species: the broad spectrum of 204 species simply cannot be interpreted. Many observers are open-minded for all birds.
Discussion: also not impressive. Reference l 225 chance the reference into: Randler et al. 2016) - What should a sentence like "The white stork is a Polish icon and is very often considered as the most Polish of all birds" mean? Phew. - l 241 - 250: too vague. - Be careful in mixing (cultural) aspects of bird "people" and aspects of general culture (e.g., role of crans in art). - Final sentence (l 259) "obsessive recreationalists" is a negative wording. I would clearly prefer "passionate". -- These are only a few selected objections.
Conclusion: Apart from my subjective impression and criticism, the work is, formally speaking, correct. A tiny brick for the large building of life sciences.

Reviewer 2 ·

Basic reporting

The manuscript is well written and clearly structured.
Relevant literature is broadly covered.
I have a few questions concerning the raw data (see general comments for the author).
The text is merely descriptive, any hypotheses are missing!
All supplemental material should be in English.

Experimental design

The manuscript covers original primary research. The research question is clear and fills a knowledge gap.
There might be a bias in the choice of the persons which received the questionnaire (see general comments for the author). Was this group really representative for Polish birdwatchers, e.g. regarding sex and age?

Validity of the findings

See comments for the author.

Additional comments

The results and disappointingly also the discussion focus very much on a few, purely descriptive aspects. Aren’t there any aspects concerning methods (e.g. what do the results mean for mapping or counting birds) or possible applications in conservation (e.g. in management or ecotourism)?

Some minor comments:
Line 53: „necessarily“ instead of „necessary“
Line 76: Different financial resources and certainly time availability will affect the circadian behaviour (which might differ from the circadian preferences). Why weren’t these aspects covered by the questionnaire?
Line 94: Choosing participants from e-mail addresses in publications only could have caused a bias since many enthusiastic birdwatchers never ever publish a single line!
Line 115: According to the methods participants were not asked for their academic degrees and jobs. Where were these derived from?
Line 143: The proportion of male respondents is pretty useless without knowing the proportion of males/females in the group of persons who received the email with the questionnaire. Accordingly the statement “that watching birds is still more popular among men than women” (line 208) is not supported by the information and data given here.
Lines 198: Detailed results (e.g. the percentages) need not to be repeated in the discussion!
Line 211: Though I appreciate the increasing trend to watch birds in females, too, the word “pleasing” is a needles appraisal.

---

## Round 0.2 · Major Revisions

Dear Piotr

As you can see, reviewer 2, who is an experienced ornithologist, is still not contend with your revision. He even suggested a rejection. But I would like to give you another chance. But we can continue only, if you are willing to make a thorough revision along the lines outlines by rev. 2.

Greetings
Michael

Reviewer 1 ·

Basic reporting

problems cleared, revision OK

Experimental design

problems cleared, revision OK

Validity of the findings

problems cleared, revision OK

Additional comments

The ms is OK now - despite some personal reservations.

Reviewer 2 ·

Basic reporting

I expected a "major revision" of this manuscript to make it of interest for a broader readership. Regrettably, only minor changes were made.

For example, I hoped to find a broader discussion on the practical value of this study in monitoring and conservation. There are heaps of literature on the pros and cons of using citizen-science data in these issues. As the authors wrote "it is important to know if their behaviour may subconsciously influence their records." How? Why don't they discuss this in more detail instead of adding just two sentences without any references to the discussion?

Experimental design

As said in my first review, I doubt that the questionaire is representative.

The authors only "... believe that this group was very representative ...., and at least covers the most active birdwatchers..." I myself know many birdwatchers personally that collect a lot of data but never have published a single line. At least I expected some discussion on this issue.

Validity of the findings

see 2

Additional comments

This is still a very descriptive manuscript with little practical or scientific value.

---

## Round 0.3 · Minor Revisions

Dear Piotr

May I ask you for a further minor revision?

Thanks and Greetings

Michael Wink
Academic Editor

Reviewer 3 ·

Basic reporting

no changes required

Experimental design

no changes required

Validity of the findings

no changes required

Additional comments

Dear Authors,
Very interesting research, I think it should be published. Despite that it is only correlative analysis, It bring a lot of new information for develop large scale monitoring scheme.
Line 21: why such research is needed, please explain
Line 78-79: as well as collecting large scale monitoring data based on birdwatchers
Line 219 - 220: please add, how many people
Line 251: and it's easy to spot especially for beginners birdswatcher
Line 280-281: Please add some references based on data from Poland

---

## Round 0.4 · accepted · Accept

Dear Piotr

Congratulations! Your revision is adequate and we can accept your contribution.

Kind regards
Michael